# Short-range order controlled amphoteric behavior of the Si dopant in Al-rich AlGaN

Igor Prozheev [1], René Bès [1], Ilja Makkonen [1], Frank Mehnke[2], Marcel Schilling[2], Tim Wernicke [2], Michael Kneissl [2,3] & Filip Tuomisto [1]

AlGaN alloys with high Al content offer the possibility to create deep ultraviolet light sources emitting at wavelengths ≤ 240 nm with enhanced quantum efficiency. However, increasing the band gap when the Al content surpasses ~80% leads to problems with $n$-type doping of AlGaN alloys with the standard choice of Si donors, due to the formation of the so-called negative Si DX center. In this paper, we show that the amphoteric nature of the Si dopant in AlGaN alloys is fundamentally controlled by the local environment and the ordering of the Ga and Al atoms in the vicinity of the Si atom. Our conclusions are based on advanced characterization sensitive to the local environment of defects and impurities, complemented by electronic structure calculations. We propose that spatial ordering of Ga and Si atoms could allow efficient $n$-type doping at even higher Al contents, including AlN.

Discovery and technological realization of alloying of III–V compound semiconductors with different band gaps ($E_g$) and stacking thin films thereof opened up the field of semiconductor optoelectronics several decades ago. The family of III-nitride semiconductors AlN ($E_g$ = 6.2 eV), GaN ($E_g$ = 3.4 eV) and InN ($E_g$ = 0.7 eV) extends the spectrum across the whole visible range and is at present the core technology in, for example, solid-state lighting. This technology is based mainly on InGaN alloy systems, while the device heterostructures typically include also thin layers of low Al content alloys. The ultra wide band gap of AlN allows band gap engineering for the development of short wavelength opto-electronic as well as power-electronic devices, such as displays with high luminosity, lasers and high-power high-frequency chips with improved stability and energy efficiency[1,2]. Moreover, synthesizing AlGaN alloys with high Al content creates the possibility to fabricate deep ultraviolet (UV) light sources emitting at wavelengths ≤240 nm with enhanced quantum efficiency[3–6]. Such UV light emitting devices can be applied, for example, in sanitizing systems against viruses and bacteria, which possibly can help to inhibit infectious diseases or provide more people with drinking water[7].

Controlling the conductivity of semiconductors is key to their functionality[8]. Already early on with conventional III–V semiconductor AlGaAs alloys, it was discovered that pushing the limits of the band gap towards higher values led to problems with $n$-type doping with the standard choice of Si as a dopant. The root cause for the inability to obtain highly conductive material was found to be the formation of the so-called negative Si DX center when the Al content surpassed ~20% in the Al$_x$Ga$_{1-x}$As alloy[9]. In this defect the Si atom normally substituting for the cation lattice site displaces strongly and undergoes a donor-to-acceptor transition[10,11]. A similar phenomenon occurs in Si-doped AlGaN alloys, albeit at much higher Al content: in AlGaN with Al content higher than 80%, the negatively charged Si DX state is formed by the axial Si-N bond rupturing along the c-axis followed by the displacement of the N atom[12–15]. Electrical compensation by the formation of cation vacancy complexes with silicon has also been suggested as the doping limiting process in high Al content AlGaN[16]. Interestingly, the DX center formation appears universal to compound semiconductor alloys with Ga/Al cations, as also in the $\beta$-(Al,Ga)$_2$O$_3$ alloys Si is an efficient donor up to a certain Al content, above which the material becomes semi-insulating irrespective of the Si doping[17]. In these oxide alloys, the bond-breaking associated with the Si DX formation depends on the site of the Si atom as the lattice hosts two inequivalent cation sites: either the Si atom or the adjacent O atoms are strongly displaced[18].

[1]Department of Physics and Helsinki Institute of Physics, University of Helsinki, P.O. Box 43, FI-00014 Helsinki, Finland. [2]Institute of Solid State Physics, Technische Universität Berlin, Hardenbergstr. 36, EW 6-1, 10623 Berlin, Germany. [3]Ferdinand-Braun-Institut (FBH), Gustav-Kirchhoff-Str. 4, 12489 Berlin, Germany. ✉e-mail: filip.tuomisto@helsinki.fi

**a** Basic alloy structure

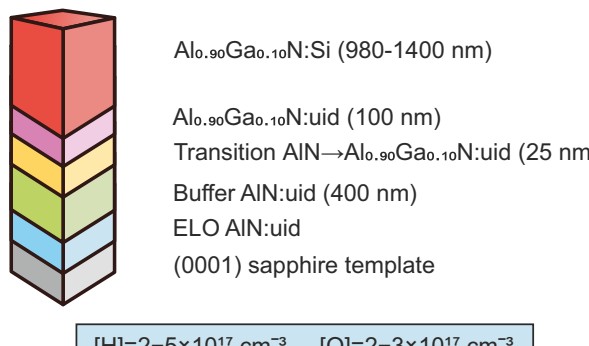

Al$_{0.90}$Ga$_{0.10}$N:Si (980-1400 nm)

Al$_{0.90}$Ga$_{0.10}$N:uid (100 nm)

Transition AlN→Al$_{0.90}$Ga$_{0.10}$N:uid (25 nm)

Buffer AlN:uid (400 nm)

ELO AlN:uid

(0001) sapphire template

[H]=2−5×10$^{17}$ cm$^{-3}$      [O]=2−3×10$^{17}$ cm$^{-3}$

**b** Resistivity as a function of Si content

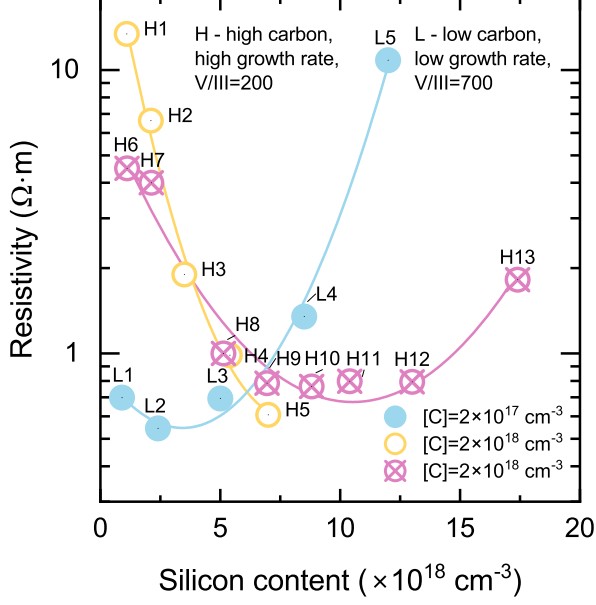

**Fig. 1 | Characteristics of the AlGaN samples. a** The basic structure (not in scale) of the samples studied is shown in the upper panel, together with the O and H concentrations. **b** Resistivity as a function of Si doping concentration, with respective C concentrations in the various sample series. The curves are drawn to guide the eye.

In this paper, we show that the amphoteric nature of the Si dopant in AlGaN alloys is fundamentally controlled by the local environment and the ordering of the Ga and Al atoms in the immediate surroundings of the Si atom. This is in contrast to the common observation in conventional III-V semiconductors, where the Si DX transformation is generally interpreted in terms of a non-local average characteristic of the material, resulting in a threshold average alloy composition as a determining factor for the phenomenon. Our conclusions are based on advanced experimental characterization sensitive to the local environment of defects and impurities on an extended set of Al$_{0.90}$Ga$_{0.10}$N epitaxial thin films doped with Si in the range from $1 \times 10^{18}$ cm$^{-3}$ to $2 \times 10^{19}$ cm$^{-3}$. Positron annihilation spectroscopy[19] reveals that negatively charged acceptor ions emerge at concentrations comparable only to the Si concentration, simultaneously with the onset of the compensating character of further increase in Si doping. X-ray absorption near edge structure spectroscopy (XANES)[20] shows that the local environment of the Si atoms changes from GaN-like to AlN-like at the same threshold concentration. We also perform electronic structure calculations[21] that show that Si is drawn to the locally Ga-rich lattice sites, retaining its donor character when these are available. As our experiments show that the presence of Ga in the vicinity of Si controls the donor character of the latter in high Al content AlGaN, we

propose that spatial ordering of these atoms could allow efficient *n*-type doping at even higher Al contents, including AlN.

## Results

### Si doping of high Al content AlGaN

The basic characteristics of the Si-doped Al$_{0.90}$Ga$_{0.10}$N epitaxial thin films discussed in this paper are shown in Fig. 1a. The samples were synthesized by metal-organic vapor phase epitaxy (MOVPE) on epitaxially laterally overgrown (ELO) AlN/sapphire, with the top layer as the material of interest[22]. The thin films were grown in two types of conditions resulting in either a relatively high carbon content ($2 \times 10^{18}$ cm$^{-3}$, samples H1 – H13), or in a relatively low carbon content ($2 \times 10^{17}$ cm$^{-3}$, samples L1 – L5), similar to ref. 23. In set H1-H5, the highest [Si] reached $7 \times 10^{18}$ cm$^{-3}$ resulting in the lowest resistivity near the possible threshold [Si] changing to the upward trend in resistivity. The set H6-H13 covers broader range of [Si] and demonstrates the predicted knee behavior of resistivity similar to samples L1-L5. We include here data on previously reported samples H1-H5[23] to highlight the consistency in both the resistivity and positron data across these two sets of samples. The growth conditions of H1-H5 are similar to H6-H13 with minor fluctuations in precursors flow. The Si doping was controlled by adjusting the SiH$_4$ flow during synthesis, resulting in the Si content ranging from $1 \times 10^{18}$ cm$^{-3}$ to $2 \times 10^{19}$ cm$^{-3}$ in the samples. The concentrations of other impurities are all below $5 \times 10^{17}$ cm$^{-3}$.

The resistivity of the AlGaN samples as a function of Si doping, shown in Fig. 1b, exhibits a similar trend in both types of synthesis conditions. First, the resistivity decreases with the increase of Si content, as expected for a dopant that should increase the free carrier concentration. Above a certain threshold, this decrease levels off and at sufficiently high Si doping concentrations the resistivity increases with further increase of Si content. This behavior where the doping efficiency is severely reduced−and even excess compensation observed−above a certain threshold Si concentration is a well known problem in *n*-type doping of AlGaN alloys[15]. With increasing Al mole fraction in AlGaN, both the Si doping efficiency and threshold Si concentration decrease, strongly limiting the achievable conductivity in high Al content AlGaN. The reduced doping efficiency with high Al contents is likely due to the higher donor activation energies and formation of Si DX centers that exist in AlN but not in GaN[14,15]. However, the reason for the vanishing and even negative doping efficiency with increasing Si concentration is not clear.

### Positron annihilation spectroscopy

Figure 2 shows the normalized (*S*, *W*) parameters measured in the Si-doped AlGaN samples at room temperature. All the measured (*S*, *W*) data are located within the area defined by the points characterizing the GaN and AlN lattices, the typical in-grown Ga vacancies in GaN (denoted by V$_{Ga}$-X), and the Al vacancy in AlN[23−26]. Our earlier work on a subset of these samples, where we performed also temperature-dependent experiments, allowed us to conclude that the data obtained in sample H1 represents the Al$_{0.90}$Ga$_{0.10}$N lattice, while the average V$_{III}$ found in Al$_{0.90}$Ga$_{0.10}$N is characterized by the data obtained in samples L1-L3[23]. The gradual shift in (*S*, *W*) parameters obtained in the samples H1 - H9 (red markers) with increasing Si content from the AlGaN lattice point towards the V$_{III}$ point indicates that the V$_{III}$ concentration in these samples increases with increasing Si concentration. The V$_{III}$ concentrations can be estimated from the positron data and are shown in Fig. 2– see the Supplementary Information (SI) for detailed analysis. It is important to note that the V$_{III}$ concentrations are an order of magnitude lower than the Si concentrations, and hence too low to account for the electrical compensation in these samples.

Interestingly, the (*S*, *W*) parameters behave very differently in samples H10−H13 and L1−L5 (blue markers). Instead of saturating at the point characteristic of V$_{III}$ that would result from further increase of the V$_{III}$ concentration with the increase of the Si content, the (*S*, *W*)

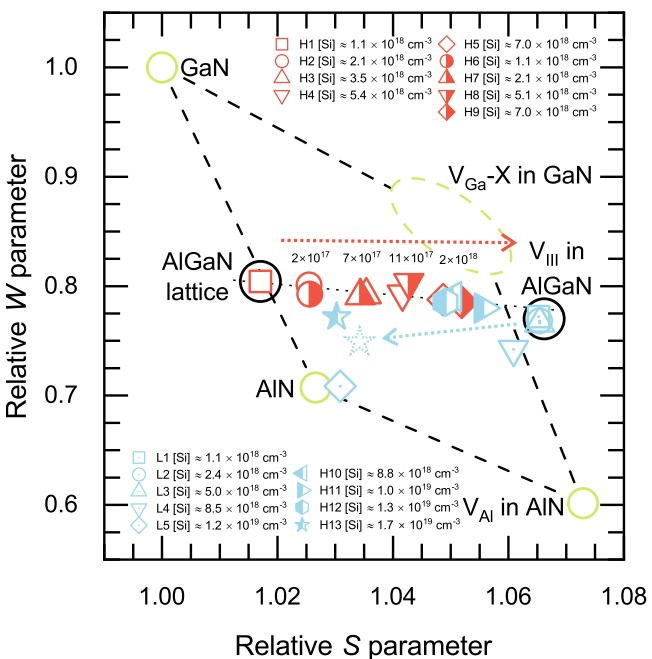

**Fig. 2 | Relative ($S$, $W$) parameters measured in the Si doped AlGaN alloy samples at room temperature.** The light green circles mark the values for the GaN lattice, AlN lattice and $V_{Al}$ in AlN, and the light green dashed ellipse shows those for in-grown $V_{Ga}$ complexed with H and O impurities or $V_N$ in GaN, as shown in earlier work[23]. The values given above the data markers show the evolution of the cation vacancy concentration with the ($S$, $W$) parameters. The data points in red/blue are from samples where the resistivity decreases/increases with increasing Si content. The arrows show the direction of increasing Si content. The dashed blue star shows where the data point for sample H13 shifts if the carbon effect is removed.

parameters shift towards the left side of Fig. 2. In the L series, the endpoint of this shift is very close to the point characteristic of the AlN lattice. In the H series the end-point $W$ parameter is close to the original AlGaN - $V_{III}$ line, but still clearly below, towards the AlN-like data of the L series. As seen in the temperature-dependent data in sample L5[23], this dramatic shift is due to the emergence of a high concentration of negative ion type defects in the samples. Detailed analysis (see SI) shows that the concentrations required for this magnitude of effect are very close to the Si concentration in these samples, and significantly higher than the concentration of carbon that also acts as a negative ion type defect for positrons, or of any other impurity. The less dramatic vertical effect in samples H10-H13 is indeed due to the higher C concentration in these samples compared to L1–L5: removing the carbon effect from the ($S$, $W$) data for sample H13 moves the endpoint to the location shown by the dashed star in Fig. 2. The most dramatic shift of the ($S$, $W$) parameters and hence the emergence of a high concentration of negatively charged non-open volume defects coincides with the total loss and reversal of the Si doping efficiency in Fig. 1b. It is also important to note that when the positron data are dominated by $V_{III}$, there is a clear over-representation of GaN-like behavior as the $V_{III}$ point is much closer to the Ga vacancies in GaN than to the Al vacancies in AlN. In contrast, when the negative ion type defects dominate the positron data the situation is reversed and the AlN-like behavior is stronger.

### X-ray absorption near edge structure spectroscopy

Figure 3a–c shows the XANES spectra obtained at the Si K-edge in selected AlGaN samples. We identify 12 characteristic features in the experimental spectra, as denoted in the figure. Features #1, #3 and #7 are so-called isosbestic points where all the spectra coincide. The

H series samples are shown in Fig. 3a, and there is a clear distinction between the samples H6–H9 and H10–H13. First, we note that the position of the whiteline (feature #2) is the same in all samples, but its intensity is clearly higher in samples H10–H13 compared to H6–H9. Second, the two peaks at 1846–1848 eV (features #4 and #5) are divided between the two sets: the #4 is clearly stronger in samples H6–H9, while #5 is clearly stronger in samples H10–H13. Additionally, sample H9 contains a small shoulder in the vicinity of feature #5, denoted as feature #6. Third, the samples H6–H9 exhibit a relatively high intensity plateau between features #8 and #9, while the samples H10–H13 exhibit two separated peaks (features #8 and #10). Fourth, and finally, the samples H6–H9 exhibit a shoulder above 1860 eV (feature #11), while samples H10–H13 show a peak at 1868 eV (feature #12). Figure 3b shows the comparison of the experimental spectra obtained in samples H13 and L2–L5. It is clear that the above-discussed features are very similar in these two sets of samples.

The experimental XANES spectra in Fig. 3b show that the local environment of Si is clearly different in samples H6–H9 compared to samples H10–H13 and L2–L5. This difference in local environment coincides with the different behavior in resistivity. Simultaneously, it coincides with the positron annihilation data resembling either GaN-rich (H6–H9) or AlN-rich environments in the AlGaN alloy.

For a quantitative assessment of the XANES spectra, we performed finite difference method near edge structure (FDMNES) simulations of Si in various environments[27]: $Si_{Ga}$ in GaN, $Si_{Al}$ in AlN, and several $Si_{Al}$ in AlN complexed with substitutional C, interstitial H, interstitial Si, and the Al vacancy with and without additional H. The simulated XANES spectra at the K-edge are shown as the total density of states vs. energy in Fig. 3c, and $Si_{Ga}$ and $Si_{Al}$ are included for comparison in Fig. 3a, b. We note that most of simulated XANES spectra show similar behavior, except for $Si_{Al}$-$H_i$ and $Si_{Al}$-$Si_i$.

In the simulated XANES spectra, interstitials near the Si atom and substitutional impurities occupying the nearest N sites contribute most significantly to the appearance of spectral features #4 and #5. Around 1847 eV, most simulated systems demonstrate common behavior, characterized by the presence of two peaks with variations in intensity and slightly also in energy. However, systems containing $Si_{Al}$-$H_i$ and $Si_{Al}$-$Si_i$ pairs deviate from this general trend. In $Si_{Al}$-$H_i$ complexes, a high-intensity peak (#4) appears below 1847 eV, followed by a smaller peak (#5) above 1847 eV, with less than one-third of the intensity of feature #4. In contrast, in the $Si_{Al}$-$Si_i$ system, feature #4 is less than 70% of the intensity of feature #5, and the latter is shifted approximately 1 eV to lower energies compared to all other samples. All simulated spectra exhibit a small peak at 1852 eV (features #7 and #8), with slight variations in centroid position and intensity.

The spectral features #9 and #10 are primarily influenced by the atomic species occupying the cation site nearest to the absorbing Si atom. For Si-complexes in AlN, all simulated spectra display a similar trend, with a peak appearing around 1860 eV, whose intensity varies depending on the composition of the system. Once again, $Si_{Al}$-$H_i$ and $Si_{Al}$-$Si_i$ systems appear as outliers. In the $Si_{Al}$-$H_i$ system, the peak shifts slightly to 1859 eV. In the $Si_{Al}$-$Si_i$ system, there are three peaks at 1859, 1861, and 1863 eV characteristic to the introduction of a Si atom. This unique behavior is exclusive to the $Si_{Al}$-$Si_i$ system, with the third peak corresponding to feature #11. For feature #12, all simulated spectra exhibit a low-intensity peak near 1867 eV. However, in the experimental spectra of H6–H9, where Si is predominantly in a Ga-rich environment, feature #12 is absent, and the spectra instead show a smooth transition to the continuum. We conclude that the slight differences between the various Si-related complexes in AlN are likely to induce broadening of the various features as observed in experiments, but they cannot account for the spectral differences between H6-H9 and each of the other groups, H10-H13 and L2-L5. We also note that the

## a Absorption at Si K-edge in H series

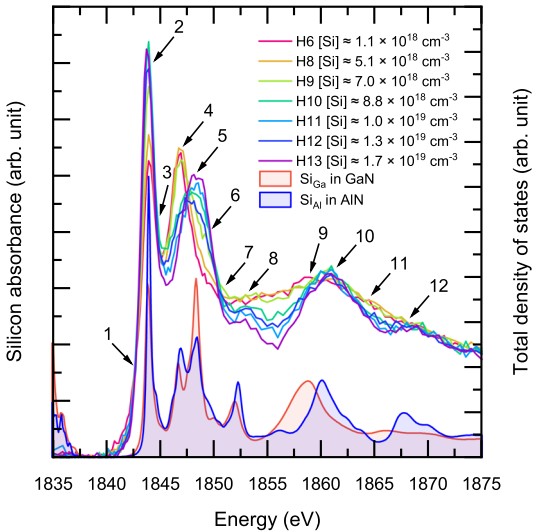

1 - isosbestic point at 1842.8 eV
2 - peak at 1844 eV
3 - isosbestic point at 1844.6 eV
4 - peak at 1846.4 eV
5 - peak at 1848.2 eV
6 - shoulder at 1849.2 eV
7 - isosbestic point at 1850.5 eV
8 - peak at 1853 eV
9 - peak at 1859 eV
10 - peak at 1861 eV
11 - shoulder at 1865 eV
12 - peak at 1868 eV

## b Absorption at Si K-edge in L series

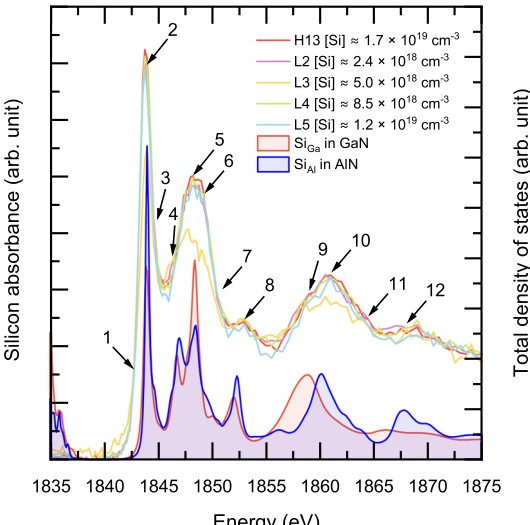

## c Simulated absorption spectra in AlN

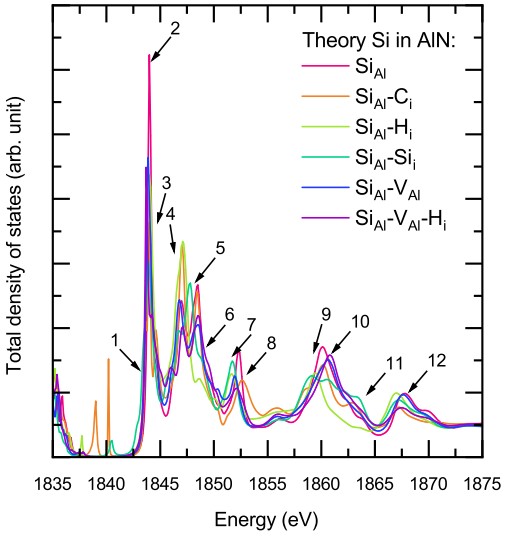

**Fig. 3 | Experimental and simulated X-ray absorption near edge spectroscopy (XANES) data at the Si K-edge. a** XANES spectra at the Si K-edge in H series Si doped AlGaN samples. **b** XANES spectra in selected H and L series Si doped AlGaN samples. Numbers 1–12 indicate characteristic isosbestic points and features.

Simulated unconvoluted XANES spectra at the Si K-edge for $Si_{Ga}$ in GaN and $Si_{Al}$ in AlN are shown for comparison. **c** Simulated unconvoluted XANES spectra of at the Si K-edge for various Si configurations in AlN.

concentrations of O, H and C in the samples are significantly lower than the Si doping.

The comparison between experiments and the two simulated spectra reveals that the whiteline has higher intensity for $Si_{Al}$ than for $Si_{Ga}$. In addition, the intensity of feature #5 is clearly stronger than that of feature #4 for $Si_{Ga}$, while there difference is not significant for $Si_{Al}$. $Si_{Ga}$ exhibits a lower intensity of the feature #8 than $Si_{Al}$, and the $Si_{Ga}$ peak at feature #9 is broad, while $Si_{Al}$ exhibits a narrower peak at feature #10 instead of #9. Finally, $Si_{Al}$ exhibits a clear peak at feature #12, while $Si_{Ga}$ shows no higher-energy peaks at this energy range, resembling feature #11. Comparing the simulations and experiments, three out of the four main observations above are in favor of the experimental XANES spectra of samples H6–H9 being dominated by cation-substituting Si in GaN-resembling local environment, and the samples H10–H13 and L2–L5 dominated by cation-substituting $Si_{Al}$ in AlN-resembling local environment. We conclude that the local environment of Si atoms is Ga-rich in samples H6–H9, and Al-rich in samples H10–H13 and L2–L5. The observed spatial correlation of Si and Ga

atoms in samples H6–H9 indicates that there is a force driving Si and Ga atoms together during synthesis.

### Electronic structure theory

State-of-the-art electronic structure calculations provide valuable insight into the electronic behavior of Si in various environments and also into the driving force behind the spatial correlation of Si and Ga atoms in the AlGaN alloys found in experiments. In binary AlN and GaN, our results on the formation of Si in the positive, neutral and stable negative DX configurations are in excellent agreement, within 0.1 eV, with previous work (see SI)[14,28]. In short, we calculate the (±) transition for the Si in AlN to be located at 0.17 eV below the conduction band minimum (CBM) that is at 6.17 eV above the valence band maximum (VBM). In GaN, the negative Si DX configuration also exists, but the (±) transition occurs 1.68 eV above the CBM.

Figure 4 shows the results of our calculations in 6 different atomic configurations of Si in AlGaN that we investigate in more detail. In all structures, we have placed 1 Si atom, 12 Ga atoms, 131 Al atoms and 144

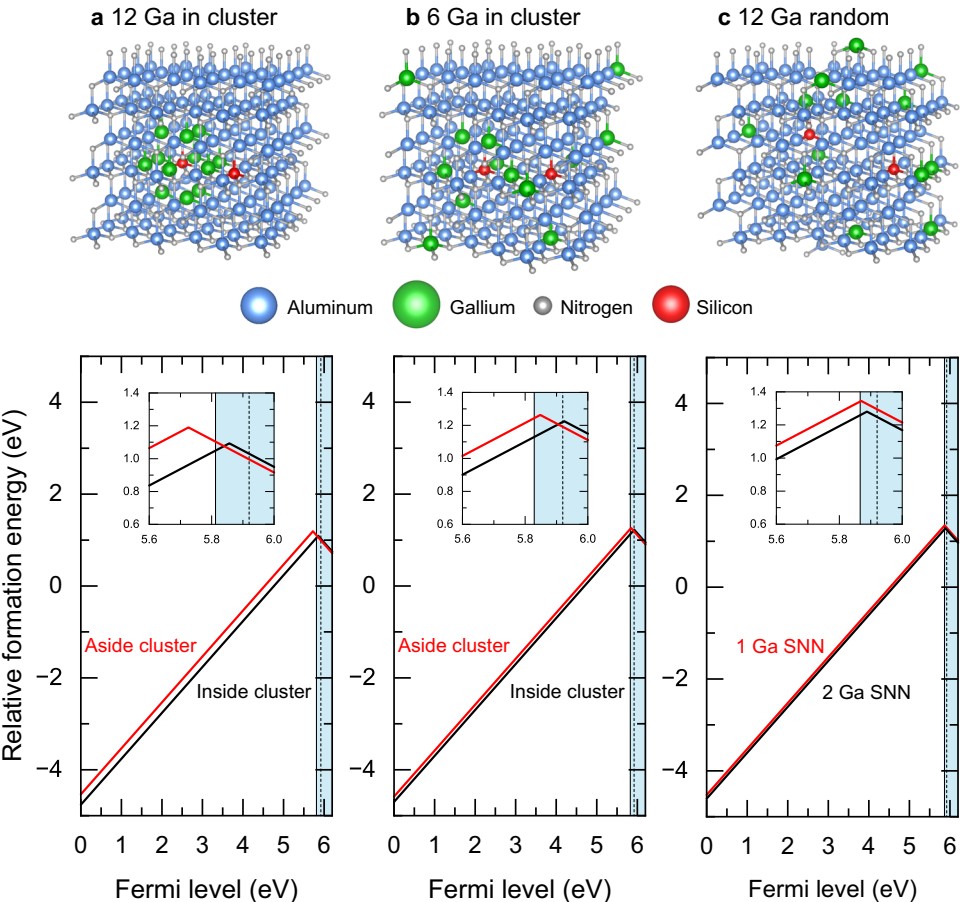

**Fig. 4 | Different supercell configurations for Si on the cation substitutional sites with varying Ga atom configurations in Al$_{131}$Ga$_{12}$N$_{144}$ and corresponding relative formation energy as a function of Fermi level.** Three different supercells are considered: (**a**) cluster of 12 Ga atoms surrounding the Al site as the second-nearest neighbor (SNN) shell, (**b**) half-cluster of 6 Ga atoms as SNN and 6 Ga atoms distributed randomly in the supercell, and (**c**) 12 Ga atoms distributed randomly in the supercell. In the formation energy plot, the solid vertical line followed by the blue shaded area at higher energies indicates the corresponding band gap calculated in each system without the Si atom. The dotted vertical line inside the blue shaded area indicates the band gap estimated for the alloy containing 90% Al. It is added to provide an idea on the variation of the calculated band gap value between different atomic configurations in the model, and it should not be interpreted as the absolute band edge position of the macroscopic alloy.

N atoms in a 288 atom supercell, corresponding to an atomic fraction of 8.3% of Ga (and 91.7% of Al). In the first system (Fig. 4a), the 12 Ga atoms fill the second-nearest-neighbor (SNN) shell of an Al site, forming an effective Ga cluster. Two different Si configurations are considered: (1) Si substituting for the central Al atom and (2) Si substituting for an Al atom next to the Ga cluster. In the second system (Fig. 4b), the SNN shell of an Al site is half-filled with 6 randomly placed Ga atoms, and the other 6 Ga atoms are distributed randomly in the rest of the supercell. Two different configurations are Si considered: (1) Si substituting for the central Al atom and (2) Si substituting for an Al atom next to the Ga half-cluster. In the third system (Fig. 4c), the 12 Ga atoms are distributed randomly in the supercell. Again, two different Si configurations are considered: (1) Si substituting for an Al atom with 2 Ga atoms as SNN and (2) Si substituting for an Al atom with 1 Ga atom as SNN. We show the formation energy of Si in these configurations that correspond to the localized Kohn-Sham states within the gap for $q = +1$ and the negatively charged Si DX configuration with a broken Si-N* bond along the c-axis.

Several observations are evident from the formation energies. First, the more there is Ga next to the Si atom in any of the configurations, the lower is the formation energy. Second, a lower number of Ga atoms in the SNN of Si results in the shift of the ($\pm$) transition level to the lower Fermi level energies within the gap, making it thermodynamically possible to form in highly *n*-type doped samples. Third,

the two different Si atomic configurations in the supercell with random Ga distribution result in very similar formation energies for both charge states (Fig 4c). Fourth, the ($\pm$) transition occurs very close to the calculated CBM in all cases.

## Discussion

Doping high Al-content AlGaN with increasing concentrations of Si, as shown in Fig. 1b, decreases the resistivity as expected up to a certain threshold concentration. Above this threshold, the resistivity increases with further increase of Si concentration. Positron annihilation results show that the concentration of compensating cation vacancy defects $V_{III}$ increases with increasing Si content (Fig. 2 and ref. 23), but the concentrations of these vacancy defects are an order of magnitude too low to efficiently compensate for the Si doping. Instead, at the highest Si doping concentrations where the resistivity increases with increasing Si content, the emergence of negatively charged acceptor-type defects with no associated open volume is evident, at concentrations comparable to the Si content. Importantly, the concentrations of the other impurities in the material, including carbon, are at least an order of magnitude lower. Hence the strong over-compensation is either due to a donor-to-acceptor transition of Si at high Si concentrations, or due to a so far unidentified intrinsic acceptor-type defect. We note that N vacancies have considerably higher formation energies than substitutional Si throughout the band gap of AlN, and hence are not likely to

play a role, even if they also exhibit a negative charge state when the Fermi level is close to the CBM[14,29].

The details of the positron annihilation data (Fig. 2) reveal that the $V_{III}$ defects that are formed at increasing concentrations with increasing Si content are characterized by a strong over-representation of a GaN-like environment. In-grown cation vacancies are typically complexed with donor-like impurities and native defects. In heavily Si doped material, it is hence likely that the $V_{III}$ defects observed by positrons are $V_{III}$–Si complexes as the concentrations of the other impurities are not high enough. As a consequence, the Si dopants that raise the Fermi level and generate the $V_{III}$ with which they are complexed are mostly in Ga-rich environments of the AlGaN alloy. In contrast, the negative acceptor defects that emerge at high Si concentrations exhibit a strong over-representation of AlN-like environments compared to the AlGaN lattice. The concentration of these negative acceptors is comparable to the Si concentration, and an order of magnitude higher than that of any other impurity. Further, the XANES spectra (Fig. 3a) show that the local environment of Si is GaN-like up to a threshold Si content, and AlN-like above that threshold. The threshold concentration depends on the carbon content and the level of pre-existing compensation that dictates the Fermi level position: lower carbon content and less compensation lead to a somewhat lower Si concentration threshold. Importantly, the threshold is the same as that observed in the positron annihilation experiments and in the electrical characteristics. In short, the experimental evidence indicates that Si is predominantly incorporated in Ga-rich environments up to a threshold Si concentration and acts as a donor. Above that concentration, Si is predominantly incorporated in AlN-like environments, and acts as a negatively charged acceptor.

The experimentally observed preferential incorporation of Si in Ga-rich environments can be understood on the basis of the results of ab initio theoretical calculations presented in Fig. 4a–c. The formation energy of Si substituting for the cation in the AlGaN lattice is lowered by the presence of Ga atoms as second nearest neighbors (SNN), and this effect is stronger the more there are Ga atoms in the 12-atom SNN shell surrounding the Si site. While the lowering is not very large, its existence provides a driving force for the local ordering of Si and Ga atoms in AlGaN. The high growth temperature, above 1000 °C, allows Si to diffuse efficiently not only on the growth surface but also when buried in the film, to get stabilized at the lowest-energy lattice sites[30]. Assuming a random alloy with no clustering or other kind of ordering, the binomial distribution of Al/Ga atoms in $Al_{0.90}Ga_{0.10}N$ leads to $2.6 \times 10^{19}$ cm$^{-3}$ cation sites with at least 50% of Ga atoms (6 out of 12) in the SNN shell. The average distance between such sites is 10–15 atomic jumps. Hence the Si diffusion that in AlN proceeds via the vacancy mechanism with an activation energy of about 3.5 eV (ref. 30) easily allows finding the Ga-rich sites. Even if the energetic difference between Ga-rich and Al-rich cation sites for Si is relatively small—roughly 0.25 eV in our calculations—this difference appears to be sufficient as seen in the experiments. We note that these observations are also in excellent agreement with the positron data showing that the cation vacancies that are most likely complexed with the Si dopants are in Ga-rich environments. Interestingly, this simple theoretical estimate of Ga-rich—defined here as at least 50% Ga atoms as SNN—lattice sites available for Si is close to the experimentally observed threshold concentration where the behavior of Si changes from donor to acceptor.

The nature of the local compositional fluctuations in a random binary alloy can be analyzed by examining the properties of the binomial distribution (see SI for details), and this provides an additional observation. We consider 200 atoms surrounding a Si atom on the cation site, similar to the size of a typical supercell in state-of-the-art electronic structure calculations. In such a case, there are 100 Al/Ga atoms surrounding the Si atom. In $Al_{0.90}Ga_{0.10}N$, the probability of finding only 0 or 1 Ga atoms among the 100 Al/Ga atoms surrounding

the Si atom on the cation site is similar to the probability of having 6 or more Ga atoms in the 12-atom SNN shell surrounding that site. It is hence possible, and even probable, that once the Ga-rich surroundings are "filled" with Si when increasing the Si concentration, a significant fraction of the "overflow" Si atoms are incorporated in surroundings that are purely AlN-like on the scale of a few nm, the size of the supercell. On this scale[31], the band structure in the vicinity of the Si atom is that of pure AlN, and the electronic behavior of Si is that found in AlN with the DX transition in the band gap. This is in good agreement with the donor activation energy being very similar in highly Si-doped AlGaN alloys with 90–95% Al content and in AlN, while for lower Al contents the Si donors are much more easily activated as the probability of having pure AlN on the few nm scale decreases very rapidly with Al content decreasing below 90%[15]. It is also important to note that these relatively large compositional fluctuations are inherent to the atomic scale, and are rapidly homogenized when the number of atoms is increased: in a 2000 atom cell that is only twice as large in diameter than the 200 atom cell, even the probability of finding a region with 95% of Al in $Al_{0.90}Ga_{0.10}N$ is vanishing (see SI).

While the lowering of Si formation energy on the cation lattice site by the addition of Ga atoms as SNN provides the driving force for the local ordering of Si and Ga atoms in AlGaN, the ordering effects on the electronic transition levels are more subtle as shown in Fig 4. It is evident that the balance between the formation energies of various charged states is delicately influenced by the surrounding Ga atoms. Also the exact position of the CBM depends on the Ga distribution in the supercell used in the calculation. It is hence extremely important to take into account local compositional fluctuations when analyzing carrier localization phenomena in AlGaN[32], similarly as recently demonstrated for InGaN alloys[31]. This is an important difference between the III-nitrides and the more conventional III–V semiconductors where the virtual crystal approximation is a reasonable approach. Our calculations show that in each of the considered Ga configurations, the (±) transition level of Si is shallower with more Ga atoms surrounding the Si site. This supports the picture obtained from experiments: the donor character of Si is the strongest when there is a maximum amount of Ga atoms surrounding it, while the probability of Si being in a negative charge state increases when the immediate surroundings are AlN-like.

Combining advanced experimental characterization and state-of-the-art electronic structure calculations allows us to show that the amphoteric behavior of Si in high Al content AlGaN alloys is controlled by short-range order with multiple Ga atoms. In addition to explaining the over-compensation at high Si doping levels, our results suggest an avenue for efficient doping by Si not only in high Al content AlGaN, but possibly also in AlN. Spatially correlating Si with multiple Ga atoms in a co-doping scheme could allow for efficient n-type doping up to much higher Si contents and carrier concentrations than so far. This could be achieved via enhanced clustering of Ga atoms in the 3D matrix, or by introducing digital superlattice structures where the Si dopants could be incorporated in the GaN or Ga-rich monolayers. Interestingly, a recent report on efficient Si doping of AlN highlights that there is excess Ga in their material[33].

## Methods
### Sample preparation
The samples were grown by metal organic vapor phase epitaxy (MOVPE) in a 3 × 2" close-coupled showerhead (CCS) Aixtron reactor. Epitaxially laterally overgrown (ELO) AlN layers were grown on (0001) c-plane sapphire substrates with a miscut of 0.1° towards the [1-100] sapphire direction[23]. The threading dislocation density in the ELO AlN/sapphire templates was $1.5 \times 10^9$ cm$^{-2}$ [22,34]. The layer structure from bottom to top consists of a AlN buffer layer (400 nm), a graded transition layer from AlN to $Al_{0.90}Ga_{0.10}N$ (25 nm), undoped $Al_{0.90}Ga_{0.10}N$

(100 nm), and a Si-doped layer of $Al_{0.90}Ga_{0.10}N$ (900–1400 nm)[35]. The composition of the samples was determined by high resolution X-ray diffractometry measuring reciprocal space maps near the (10−15) AlN reflex under consideration of the layer strain state[4,36]. The resistivity of the samples was determined by contactless (eddy current) resistivity measurements in a Delcom system. Impurity concentrations (Si, C, O, H) were determined by secondary ion mass spectrometry by Evans Analytical group. In H series, the impurities contents were $[H] = 5 \times 10^{17}\ cm^{-3}$, $[O] = 3 \times 10^{17}\ cm^{-3}$ and $[C] = 2 \times 10^{18}\ cm^{-3}$. In L series, the impurities contents were $[H] = 2 \times 10^{17}\ cm^{-3}$, $[O] = 2 \times 10^{17}\ cm^{-3}$ and $[C] = 2 \times 10^{17}\ cm^{-3}$.

## Positron annihilation spectroscopy

The Doppler broadening of the positron-electron annihilation radiation was recorded with a slow positron beam at varied energies in the range of 0.5–25 keV at room temperature in all samples[19,37]. A high purity germanium detector with energy resolution of 1.27 keV at the 511 keV annihilation line was used to collect $10^6$ counts in the annihilation spectra. Conventional $S$ and $W$ parameters determined at positron implantation energies corresponding to the layer of interest were used to estimate the concentrations of negative and neutral cation vacancies in the Si-doped AlGaN samples. The $S$ parameter is defined as the fraction of counts around the ≤0.96 keV (0.4 a.u.) central region of the peak, and the $W$ parameter is defined in the tail of the peak in the energy range of (3.00–7.60 keV) (1.6–4.0 a.u.) from the center. The measured ($S$, $W$) parameters are shown in this paper as normalized to those obtained in a $p$-type GaN reference sample, representing the GaN lattice[24].

The concentrations $c_V$ of the cation vacancies are determined from the trapping rate $\kappa_V = \mu_V c_V$ estimated by analyzing the $S$ ($W$) parameter data, where $\mu_V = 3 \times 10^{15}\ s^{-1}$ is the trapping coefficient at the vacancy. Importantly, we take into account the effect of negative ion defects that act as shallow traps for positrons and are efficient in positron trapping even at room temperature in AlN and Al-rich AlGaN[23]. When this is the case, the measured $S$ parameter depends on the parameters of the lattice ($S_B$) and the vacancy ($S_V$) in the following way (a similar equation holds for the $W$ parameter):

$$S = \frac{\lambda_B + \kappa_{ion}}{\lambda_B + \kappa_V + \kappa_{ion}} S_B + \frac{\kappa_V}{\lambda_B + \kappa_V + \kappa_{ion}} S_V. \tag{1}$$

The negative ions whose trapping rate is denoted with $\kappa_{ion}$ produce the same annihilation parameters as the lattice. Here, $\lambda_B$ is the annihilation rate in the lattice, and it is the inverse of the positron lifetime in the lattice $\tau_B = \lambda_B^{-1} = 160$ ps in GaN/AlN[23]. The trapping coefficient of the negative ions is the same as that of the cation vacancies[19,37,38].

## X-ray absorption spectroscopy

We acquired fluorescence yield spectra in X-ray absorption near-edge structure region at Si K-edge (-1839 eV) in the selected samples (L2−L5 and H6−H13) on the LUCIA beamline at the SOLEIL synchrotron facility[39]. The 2.75 GeV beam was operating in a hybrid refill mode at 450 mA. The energy calibration was performed with a InSb(111) monochromator using the first inflection point of a Si reference spectrum, and fixed at 1841 eV. Acquisitions were made in the continuous scan mode from 40 eV below the silicon absorption edge (-1839 eV) up to 60 eV above. The varying energy steps were 1 eV, 0.2 eV, 0.5 eV and 1 eV in the corresponding energy regions 1800–1836 eV, 1836.2–1850 eV, 1850.5–1880 eV and 1881–1900 eV, respectively. With the counting time of 10 s on each point, one scan recording time sums up to 2280 s. We acquired at least 3 scans to obtain a suitable signal to noise ratio.

## Finite difference method near edge structure calculations

We used Finite Difference Method Near Edge Structure (FDMNES) software to obtain total electron density of state of 9 GaN:Si and 10 AlN:Si systems[27,40]. The self-consistent calculations were performed with Finite Difference Method in the range 30 eV below and 80 eV above the Si K-edge with a step of 0.1 eV. We put from one to three absorbing Si atoms on metal and interstitial sites in the center of wurtzite AlN or GaN supercells and varied the first nearest neighbors of Si absorbers with different interstitial impurities. Prior to simulations we performed ionic relaxation of all systems with conjugate gradient algorithm and Projector Augmented Wave Perdew-Burke-Ernzerhof (PAW PBE) functional on VASP 5.4[41–45]. The final state and potential calculations were performed with radius values of 7 and 8 Å, respectively.

## Electronic structure theory

Defect formation energy calculations were performed with density functional theory (DFT) using VASP 5.4 code. In the calculations, we applied the projector augmented-wave (PAW) method to Perdew-Burke-Ernzerhof (PBE) potentials from 04 January 2001, 08 April 2002, 08 April 2002, and 05 January 2001 for Al, N, Ga, and Si, respectively. The 10 3$d$ electrons of Ga were included in the core shell. All the calculations were spin-polarized. We used lattice parameters corresponding to the wurtzite phases of AlN ($a = b = 3.11$ Å, $c/a = 1.60$) and GaN ($a = b = 3.20$ Å, $c/a = 1.62$)[14], and cut-off energy of 400 eV. The orthorhombic primitive cell contained 8 atoms, which was further scaled up to super cells containing 96 and 288 atoms. We performed optimization for lattice parameters $a$ and $c$, plane wave cut-off energy and $k$-points meshes, and full ionic relaxation for the reference cells. We used the range-separated hybrid functional HSE06 which mixes the Hartree-Fock (HF) and generalized gradient approximation (GGA) parametrized by PBE. The fraction of exchange in a Hartree-Fock-type calculations was set to $\alpha = 0.33$ for AlN and AlGaN alloys, and $\alpha = 0.31$ for GaN. We used the default HF screening factor. We obtained band gap values of 6.17 and 3.58 eV for AlN:$Si_{Al}$ and GaN:$Si_{Ga}$ systems containing 96 atoms and calculated with $3 \times 3 \times 3$ Gamma-centered $k$-points mesh. While our results agree with the previously reported values[28,46], minor variations are due to the use of different $k$-point meshes and potentials. The use of $k$-points mesh containing Gamma point was tested for convergence against denser Gamma-centered $k$-points meshes in 288 atoms systems. We further used $1 \times 1 \times 1$ Gamma-centered $k$-point mesh to reduce the computational cost for larger systems. We calculated the formation energy of a charged defect as the energy difference between the investigated system and the components in their reference states following the procedure outlined in the reviews by Van de Walle and Neugebauer[47], and Freysoldt et al.[21], originally formulated by Zhang and Northrup[48]. For $Si_{III}^0$ and $Si_{III}^+$, Si occupies the metallic cation site of the corresponding lattice, with relaxations of the nearest-neighbor nitrogen atoms. For $Si_{III}^-$, the lowest energy DX configuration was achieved with the broken Si-N axial bond along the $c$ axis and large N atom displacement[14]. The charged states of the studied defects were modeled by changing the corresponding amount of electrons in the system. The partial density of states of introduced levels was calculated in order to determine whether the electron is localized. We determine the electrostatic local potential and dielectric constant tensors for AlN and GaN to apply finite size corrections for charged-defects[49]. The DFT results shown in Fig. 4a–c are aligned on a relative scale for direct comparison within the same atomic configuration by aligning the electrostatic potentials between the cells with and without a defect. We neglected the growth conditions (metal- or nitrogen-rich) of the systems and left the formation energy in the relative scale. Taking the chemical potentials into account will move all the lines at the same time along the vertical axis to the absolute values in relation to the specified growth condition. We note that the Si DX is metastable in all the studied cases, with the

simple $Si_{III}^{-}$ configuration having a total energy lower by 0.15–0.50 eV, but the electron is not localized in this configuration and we present the Si DX data in the figures. We also checked the effect of the Ga atoms as nearest-neighbors of the displaced N atom in the Si DX configuration in the random supercell[50], and found no difference to the other configurations in this supercell (the data are not shown for the sake of clarity).

## Data availability

Source data for Figs. 1–4 in this study are provided with the paper. Source data are provided with this paper.

## Code availability

All DFT calculations were performed with VASP, which is proprietary software for which the Tuomisto lab owns a license.

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

## Acknowledgements

This material is based upon work supported by the Air Force Office of Scientific Research under award number FA8655-23-1-7057 (F.T.). This work has been partially supported by Finnish Cultural Foundation through a personal grant (I.P.), through the Additional Million-euro Funding to Science programme grant 00231172 (F.T.), and supported by the German Federal Ministry of Education and Research (BMBF) within the "Advanced UV for Life" consortium (M.K.). We acknowledge the Proposal Review Committee of SOLEIL for provision of their synchrotron radiation facilities (proposal no. 20210395) and beamtime allocation on the LUCIA beamline (R.B.). The authors wish to acknowledge CSC - IT Center for Science, Finland, for generous computational resources (project 2000028) (I.M.). The authors wish to thank S. Hagedorn, Ferdinand-Braun-Institut (FBH), Berlin, for providing the ELO AlN/sapphire templates, P. Desgardin, Université Orléans, Orléans, France, for help with Doppler experiments, beamline scientist N. Trcera, LUCIA beamline, SOLEIL Synchrotron, Saint-Aubin, France, for technical assistance, and Y. Joly, Institut Néel, CNRS, Grenoble, France, and J. L. Lyons, US Naval Research Laboratory, Washington DC, USA, for fruitful discussions.

## Author contributions

I.P. with F.T. designed the study. F.M., M.S., T.W. and M.K. supplied samples and characterization with SIMS, XRD and resistivity measurements. I.P. performed the positron annihilation experiments. I.P. performed the X-ray absorption near edge structure experiments and FDMNES calculations with help of R.B. I.P. performed PAW-DFT calculations with help of I.M. I.P., R.B., I.M., and F.T. analyzed the results. All authors reviewed the manuscript.

## Competing interests

The authors declare no competing interests.
