## [Transparent Peer Review file · Nature Communications]

Short-range order controlled amphoteric behavior of the Si dopant in Al-rich AlGaN

Corresponding Author: Professor Filip Tuomisto

Version 0:

Reviewer comments:

Reviewer #1

(Remarks to the Author)

This work is particularly convincing and it is relevant for the understanding of the operating conditions of very specialized devices based on AlGaN alloys. This alone would not justify an acceptance from my point of view. However it has been recently demonstrated the observation of persistent photoconductivity (PPC) in boron nitride epilayers grown by MOCVD and NOT in bulk crystals. MOCVD grown BN epilayers are contaminated by silicon and carbon in that case. The origin of this PPC is still very mysterious as nothing is really known about the behaviors of Si and C in the last of the nitrides: BN. I believe this work is very helpful for the future understanding the conductivity of BN crystals and/or epilayers. Therefore I recommend its publication in a journal of large scale reading. Nature communications is the ad hoc target for such purpose.

Reviewer #2

(Remarks to the Author)

The paper investigates the energetic, electronic, and structural properties of Si dopants in AlGaN alloys. The authors combine positron annihilation spectroscopy, XANES (supported by the difference method near-edge structure), and DFT calculations to investigate and provide insights into the amphoteric behavior of substitutional Si. Their key finding concerns the origin of the amphoteric behavior of substitutional Si, which they attribute to Si incorporation occurring preferentially in locally Ga-rich regions of the alloy. The term "short-range order," used by the authors to describe the high Ga content resulting from random alloy fluctuations, is not be appropriate.

The experimental evidence supports their main conclusion: a significantly lower concentration of compensating defects and the predominant incorporation of Si at locally Ga-rich sites, provided the Si concentration remains below a certain threshold. However, the supporting DFT calculations require further clarification:

1. The +/- transition level of substitutional Si shifts deeper relative to the conduction band minimum CBM of Al_{0.9}Ga_{0.1}N as the number of Ga atoms in the second-nearest-neighbor shell of Si increases. The authors claim that the position of this level with respect to the CBM of the Al₁₃₂Ga₁₂N₁₄₄ supercells is relevant. However, it is unclear how they align the electronic structures (band edges) of supercells with different alloy compositions. Moreover according to their calculations and interpretation a local composition of Al₁₃₂Ga₁₂N₁₄₄ in an Al_{0.9}Ga_{0.1}N alloy should create a ~0.1 eV deep QW (or QD) for electrons. The authors should further elaborate on this aspect as well.
2. The authors argue that at high temperatures, Si atoms are sufficiently mobile to diffuse and incorporate into Ga-rich sites, despite a rather high diffusion barrier of 3.5 eV. While this explanation appears reasonable, the formation energy difference between Si incorporation at Ga-rich and Ga-poor sites is less than 0.2 eV, comparable to k_B*T. Although high temperatures facilitate diffusion, they also make entropy contributions significant. This contrancticts their interpretation of preferential incorporation at locally Ga rich regions.
3. The discussion of compositional fluctuations in 200-atom supercells, which are typical for DFT calculations, lacks clarity.

Additional Comments:

- Figure 4 (Supplementary Material): The caption should not state "Atomic and electronic structure [...]" as it only presents atomic structure and a charge density plot. "Electronic structure" refers to something different.
- Page 6, line 142: The text states, "we have placed 1 Si atom, 12 Ga atoms, 132 Al atoms, and 144 N atoms [...]". This is a typo. It should be (for substitutional Si) 1 Si, 12 Ga, and 131 Al atoms.

Reviewer #3

(Remarks to the Author)

In this manuscript, the behavior of Si in high-Al-content (Al > 80%) AlGa_N was studied to overcome the n-type doping problem, which is caused by the formation of the negative Si DX center. The authors suggest how the amphoteric nature of the impurity Si can be controlled using experimental techniques such as positron annihilation spectroscopy and X-ray absorption near edge structure spectroscopy combined with theoretical calculations. Their findings indicate that the amphoteric behavior of Si is strongly affected by the local environment, particularly the short-range order involving multiple Ga atoms. Based on these findings, the authors suggest a pathway for more effective Si doping in high-Al-content AlGa_N and/or AlN.

This work is of significant interest to the III-nitride research community as it provides valuable insights into the limited conductivity problem observed in high-Al-content AlGa_N alloys, which are crucial for deep-UV light sources, such as light-emitting or laser diodes. The study combines high-quality experiments with careful analyses, supported and validated by numerical simulations. I would recommend this manuscript for publication after the authors address the following comments.

1. Figure 1: The resistivity data has two groups for H samples: H1-H5 and H6-H13. What distinguishes these two groups? It seems likely that H1-H5 correspond to the samples from Ref. 23, while H6-H13 are newly measured in this study. Is it correct? Given the differences in behavior between these two sets, additional explanation is necessary.
2. l. 99: The isosbestic point is not #2, but #3. #2 corresponds to the white line.
3. l. 107: The text states, "Figure 3b shows the comparison of the experimental spectra obtained in samples H10 – H13 and L2 – L5." However, it seems that only H13 is shown from the H series.
4. l. 117: The manuscript states, "We note that there are no major differences in the simulated XANES spectra between the various Si_{Al} complexes, except for Si_{Al}-Si_i." However, if Si_{Al}-Si_i exhibits a major major difference, the same could be argued for Si_{Al}-H_i. While I agree with the statement that "these complexes are not likely to be present at important concentrations" (l. 120) and they may not be important, a more detailed analysis of the spectral differences would be beneficial.
5. l. 142: 132 Al atoms should be 131 Al atoms assuming Si substitutes Al. Please verify.
6. l. 170: Typographic error: t othe -> to the
7. How were the formation energies calculated? Was the usual Zhang-Northrup type formula used? Please clarify.
8. l. 290: Was a Gamma-point-only sampling sufficient for convergence in the larger system? Has the convergence been checked? Given that the (+/-) transition levels are located very close to the conduction band minimum, even a subtle change in formation energies could be important.
9. l. 296: terminology "polarity": The term polarity appears to be used in a context related to growth conditions. If this is the intended meaning, "polarity" may not be the most appropriate term. Please reconsider.
10. Figure 4: The DX center appears unstable except in the case of 12 Ga in cluster Aside Cluster case. The (+/-) transition level seems to be above the gap, assuming that the band gap position is at the vertical dotted line. Have you calculated the effective correlation energy U for these states? Additionally, in the supplementary information l. 66, U is used without definition. This should be explicitly defined.

Version 1:

Reviewer comments:

Reviewer #1

(Remarks to the Author)

I am still convinced that this paper deserves publication in nature communications specifically after the authors have now improved the manuscript with ad-hoc answers to my other colleagues referees #2 and #3 in order to consolidate the presentation.

I have also the feeling that using SRO is appropriate for describing the geometrical situation around silicon. My recommendation is therefore again to accept.

Reviewer #2

(Remarks to the Author)

The authors have successfully addressed most of the reviewers' comments.

However, I still have a couple of points that require clarification:

1. Ga-rich regions due to random alloy fluctuations do not constitute short-range order. These regions are randomly distributed within the alloy and arise from statistical fluctuations. For instance, would such Ga-rich regions—emerging in an otherwise random alloy configuration—be reflected in the Warren-Cowley short-range order parameters?
2. Band alignment can be achieved or estimated by aligning w.r.t. the branch-point energy.

Reviewer #3

(Remarks to the Author)

The authors diligently addressed all the provided comments, and the manuscript has been sufficiently improved. I now recommend it for publication.

Manuscript: NCOMMS-24-80259

Title: Short-range order controlled amphoteric behavior of the Si dopant in Al-rich AlGa_N

Numbered citations refer to the numbering in the revised manuscript. The modified parts of the manuscript are highlighted with red color. Comments by reviewers in blue color, and our answers are in black color.

Reviewer #1 (Remarks to the Author):

R1: This work is particularly convincing and it is relevant for the understanding of the operating conditions of very specialized devices based on AlGa_N alloys. This alone would not justify an acceptance from my point of view.

However it has been recently demonstrated the observation of persistent photoconductivity (PPC) in boron nitride epilayers grown by MOCVD and NOT in bulk crystals. MOCVD grown BN epilayers are contaminated by silicon and carbon in that case.

The origin of this PPC is still very mysterious as nothing is really known about the behaviors of Si and C in the last of the nitrides: BN.

I believe this work is very helpful for the future understanding the conductivity of BN crystals and/or epilayers.

Therefore I recommend its publication in a journal of large scale reading. Nature communications is the ad hoc target for such purpose.

A: We agree with the point made by the reviewer about the importance of the persistent photoconductivity in boron nitride epilayers grown by MOCVD. Indeed, discussion of the role and amphoteric behaviour of Si in AlGa_N alloys can help in understanding the roles of Si and C in studying defect-related phenomena in BN epilayers and crystals.

Reviewer #2 (Remarks to the Author):

R2: The paper investigates the energetic, electronic, and structural properties of Si dopants in AlGa_N alloys. The authors combine positron annihilation spectroscopy, XANES (supported by the difference method near-edge structure), and DFT calculations to investigate and provide insights into the amphoteric behavior of substitutional Si. Their key finding concerns the origin of the amphoteric behavior of substitutional Si, which they attribute to Si incorporation occurring preferentially in locally Ga-rich regions of the alloy. The term "short-range order," used by the authors to describe the high Ga content resulting from random alloy fluctuations, is not be appropriate.

A: We used the term "short-range order" to refer to the immediate local environment of Si atoms within the range of second nearest neighbour shell to be the determining factor for donor/acceptor behavior of Si which is directly evident from the differences in crystalline lattice around Si from the XANES experiments. Our opinion is that the use of this term is appropriate: short range order (SRO for short) in crystalline alloys refers

to the regular and predictable arrangement of atomic species over a short distance, usually with one or two atom spacings.

- R2: The experimental evidence supports their main conclusion: a significantly lower concentration of compensating defects and the predominant incorporation of Si at locally Ga-rich sites, provided the Si concentration remains below a certain threshold. However, the supporting DFT calculations require further clarification:

The +/- transition level of substitutional Si shifts deeper relative to the conduction band minimum CBM of Al_{0.9}Ga_{0.1}N as the number of Ga atoms in the second-nearest-neighbor shell of Si increases. The authors claim that the position of this level with respect to the CBM of the Al₁₃₂Ga₁₂N₁₄₄ supercells is relevant. However, it is unclear how they align the electronic structures (band edges) of supercells with different alloy compositions. Moreover according to their calculations and interpretation a local composition of Al₁₃₂Ga₁₂N₁₄₄ in an Al_{0.9}Ga_{0.1}N alloy should create a ~0.1 eV deep QW (or QD) for electrons. The authors should further elaborate on this aspect as well.

- R2: The experimental evidence supports their main conclusion: a significantly lower concentration of compensating defects and the predominant incorporation of Si at locally Ga-rich sites, provided the Si concentration remains below a certain threshold. However, the supporting DFT calculations require further clarification:

The +/- transition level of substitutional Si shifts deeper relative to the conduction band minimum CBM of Al_{0.9}Ga_{0.1}N as the number of Ga atoms in the second-nearest-neighbor shell of Si increases. The authors claim that the position of this level with respect to the CBM of the Al₁₃₂Ga₁₂N₁₄₄ supercells is relevant. However, it is unclear how they align the electronic structures (band edges) of supercells with different alloy compositions. Moreover according to their calculations and interpretation a local composition of Al₁₃₂Ga₁₂N₁₄₄ in an Al_{0.9}Ga_{0.1}N alloy should create a ~0.1 eV deep QW (or QD) for electrons. The authors should further elaborate on this aspect as well.

- A: This is a good point, that the comparison of different alloy compositions would require the alignment of electronic structures. Several steps are required for this in case of alloy models with different arrangements of atoms: (1) determining the alignment between the average bulk electrostatic potential and the band edges in a bulk semiconductor calculation, (2) determining the average electrostatic potential inside the slab, and in the vacuum in slab-vacuum interface calculation, (3) aligning the band edges with respect to vacuum through the average electrostatic potential. For a high number of structurally varied supercells this would be an extremely costly calculation, and still with some degree of uncertainty due to the averaging of the electrostatic potential. We avoid this by comparing the results within the same atomic configuration by aligning the

electrostatic potentials between the cells with and without defects (as implemented in the electrostatic correction by Freysoldt *et al.*). This means that DFT results shown in Figs. 4 a-c are aligned on a relative scale within the same system for direct comparison. We have made this clear in the manuscript by adding the following text to the section on electronic structure theory:

“The DFT results shown in Figs. 4a-c are aligned on a relative scale for direct comparison within the same atomic configuration by aligning the electrostatic potentials between the cells with and without a defect.”

We understand that by the 0.1 eV quantum well or quantum dot the reviewer is referring to the energy difference between the dotted vertical lines and the blue shaded areas in Figure 4. There, the dotted vertical line denotes the band edge position and band gap value calculated for the alloy model in question without Si. The left edge of the blue shaded area marks the band gap value interpolated from GaN and AlN gaps assuming 90% Al concentration. We now see that drawing the figure in this way may lead to the interpretation suggested by the reviewer, and have modified it to make the situation more clear. Importantly, the band edges of the interpolated 90% AlGa_{0.1}N alloys are not aligned with respect to the band edges of the models in Figs. 4a-c. Based on our calculations, we cannot speculate on the possibility that the Ga-rich regions could trap electrons.

In order to clarify this point and avoid potential misunderstandings, we have modified the figure so that the band edge resulting from our calculations for each of the models in Figs. 4a-c is denoted by a solid vertical line, followed with a shaded region covering the higher energies (that, representing the conduction band). The interpolated value for 90% AlGa_{0.1}N is shown as a dotted vertical line in the shaded region. We have also modified the caption of Fig. 4 to read:

“The solid vertical line followed by the blue shaded area at higher energies indicates the corresponding band gap calculated in each system without the Si atom. The dotted vertical line inside the blue shaded area indicates the band gap estimated for the alloy containing 90% Al. It is added to provide an idea on the variation of the calculated band gap value between different atomic configurations in the model, and it should not be interpreted as the absolute band edge position of the macroscopic alloy.”

- R2: The authors argue that at high temperatures, Si atoms are sufficiently mobile to diffuse and incorporate into Ga-rich sites, despite a rather high diffusion barrier of 3.5 eV. While this explanation appears reasonable, the formation energy difference between Si incorporation at Ga-rich and Ga-poor sites is less than 0.2 eV, comparable to $k_B \cdot T$. Although high temperatures facilitate diffusion, they also make entropy contributions significant. This contradicts their interpretation of preferential incorporation at locally Ga rich regions.

A: We agree that the energetic difference between Ga-rich and Ga-poor sites is small. However, we are suggesting an explanation to the observed experimental results rather than developing a thermodynamic model that would describe in detail the behavior of Si atoms at out of equilibrium conditions. Still, at elevated temperatures Si atoms are sufficiently mobile to diffuse to Ga-rich sites where they are bound to stay due to the increasing effect of the high diffusion barrier during the cooling phase from high temperature to room temperature when the entropy terms decreases. We have clarified the text on this aspect to read by rewriting the paragraph discussing this aspect in the following way (the paragraph starting with “The *ab initio* theoretical calculations...” in the original manuscript):

“The experimentally observed preferential incorporation of Si in Ga-rich environments can be understood on the basis of the results of *ab initio* theoretical calculations presented in Fig. 4. The formation energy of Si substituting for the cation in the AlGaN lattice is lowered by the presence of Ga atoms as second nearest neighbors (SNN), and this effect is stronger the more there are Ga atoms in the 12-atom SNN shell surrounding the Si site. While the lowering is not very large, its existence provides a driving force for the local ordering of Si and Ga atoms in AlGaN. The high growth temperature, above 1000 °C, allows Si to diffuse efficiently not only on the growth surface but also when buried in the film, to get stabilized at the lowest-energy lattice sites.[30] Assuming a random alloy with no clustering or other kind of ordering, the binomial distribution of Al/Ga atoms in Al_{0.90}Ga_{0.10}N leads to $2.6 \times 10^{19} \text{ cm}^{-3}$ cation sites with at least 50% of Ga atoms (6 out of 12) in the SNN shell. The average distance between such sites is 10-15 atomic jumps. Hence the Si diffusion that in AlN proceeds via the vacancy mechanism with an activation energy of about 3.5 eV (Ref. [30]) easily allows finding the Ga-rich sites. Even if the energetic difference between Ga-rich and Al-rich cation sites for Si is relatively small – roughly 0.25 eV in our calculations – this difference appears to be sufficient as seen in the experiments. We note that these observations are also in excellent agreement with the positron data showing that the cation vacancies that are most likely complexed with the Si dopants are in Ga-rich environments. Interestingly, this simple theoretical estimate of Ga-rich – defined here as at least 50% Ga atoms as SNN – lattice sites available for Si is close to the experimentally observed threshold concentration where the behavior of Si changes from donor to acceptor.”

R2. The discussion of compositional fluctuations in 200-atom supercells, which are typical for DFT calculations, lacks clarity.

A: This paragraph is indeed quite complicated and relies heavily on the data and discussion in the Supporting Information. We have improved the text to increase the clarity by rewriting this paragraph (starting with “The local compositional fluctuations...” in the original manuscript) in the following way:

“The nature of the local compositional fluctuations in a random binary alloy can be analyzed by examining the properties of the binomial distribution (see SI for details), and this provides an additional observation. We consider 200 atoms surrounding a Si atom on the cation site, similar to the size of a typical supercell in state-of-the-art electronic structure calculations. In such a case, there are 100 Al/Ga atoms surrounding the Si atom. In $\text{Al}_{0.90}\text{Ga}_{0.10}\text{N}$, the probability of finding only 0 or 1 Ga atoms among the 100 Al/Ga atoms surrounding the Si atom on the cation site is similar to the probability of having 6 or more Ga atoms in the 12-atom SNN shell surrounding that site. It is hence possible, and even probable, that once the Ga-rich surroundings are "filled" with Si when increasing the Si concentration, a significant fraction of the "overflow" Si atoms are incorporated in surroundings that are purely AlN-like on the scale of a few nm, the size of the supercell. On this scale [31], the band structure in the vicinity of the Si atom is that of pure AlN, and the electronic behavior of Si is that found in AlN with the DX transition in the band gap. This is in good agreement with the donor activation energy being very similar in highly Si-doped AlGaN alloys with 90-95% Al content and in AlN, while for lower Al contents the Si donors are much more easily activated as the probability of having pure AlN on the few nm scale decreases very rapidly with Al content decreasing below 90%. [15] It is also important to note that these relatively large compositional fluctuations are inherent on the atomic scale, and are rapidly homogenized when the number of atoms is increased: in a 2000 atom cell that is only twice as large in diameter than the 200 atom cell, even the probability of finding a region with 95% of Al in $\text{Al}_{0.90}\text{Ga}_{0.10}\text{N}$ is vanishing (see SI).”

R2: Figure 4 (Supplementary Material): The caption should not state “Atomic and electronic structure [...]” as it only presents atomic structure and a charge density plot. "Electronic structure" refers to something different.

A: Good point, we corrected the caption to read:

“Atomic structure and visualized charge density in the Si_{Al}^- configuration along (a) a- and (b) c-axis.”

R2: Page 6, line 142: The text states, “we have placed 1 Si atom, 12 Ga atoms, 132 Al atoms, and 144 N atoms [...]”. This is a typo. It should be (for substitutional Si) 1 Si, 12 Ga, and 131 Al atoms.

A: We thank the referee for spotting this typographical error, it was corrected to 131.

Reviewer #3 (Remarks to the Author):

R3: In this manuscript, the behavior of Si in high-Al-content ($\text{Al} > 80\%$) AlGaN was studied to overcome the n-type doping problem, which is caused by the formation of the negative Si DX center. The authors suggest how the amphoteric nature of the impurity Si can be controlled using experimental techniques such as positron annihilation

spectroscopy and X-ray absorption near edge structure spectroscopy combined with theoretical calculations. Their findings indicate that the amphoteric behavior of Si is strongly affected by the local environment, particularly the short-range order involving multiple Ga atoms. Based on these findings, the authors suggest a pathway for more effective Si doping in high-Al-content AlGa_N and/or AlN.

This work is of significant interest to the III-nitride research community as it provides valuable insights into the limited conductivity problem observed in high-Al-content AlGa_N alloys, which are crucial for deep-UV light sources, such as light-emitting or laser diodes. The study combines high-quality experiments with careful analyses, supported and validated by numerical simulations. I would recommend this manuscript for publication after the authors address the following comments.

A: We thank the reviewer for the very positive attitude towards our work.

1. Figure 1: The resistivity data has two groups for H samples: H1-H5 and H6-H13. What distinguishes these two groups? It seems likely that H1-H5 correspond to the samples from Ref. 23, while H6-H13 are newly measured in this study. Is it correct? Given the differences in behavior between these two sets, additional explanation is necessary.

A: Indeed, the group H1-H5 corresponds to the previously reported data [23]. These samples were grown at low V/III ratio with high C content. The highest silicon doping in samples H1-H5 reached $7 \times 10^{18} \text{ cm}^{-3}$ resulting in the lowest resistivity values. We hypothesized that further increase in [Si] in H series might lead to the knee behaviour observed in L series. Hence, a new set of samples H6-H13 was grown with [Si] ranging between $1 \times 10^{18} - 2 \times 10^{19} \text{ cm}^{-3}$. The growth conditions of H1-H5 are similar to H6-H13 with minor fluctuations in precursors flow. The samples H6-H13 are newly measured in this study. We still find it necessary to include the previously reported data on the H1-H5 samples to show the consistency in both the resistivity and positron data across these two sets of samples.

To clarify this aspect, we have modified the manuscript in the section on Si doping of high Al content AlGa_N in the main manuscript by adding the following text:

“In set H1-H5, the highest [Si] is $7 \times 10^{18} \text{ cm}^{-3}$ resulting in the lowest resistivity near the possible threshold [Si] changing to the upward trend in resistivity. The set H6-H13 covers broader range of [Si] and demonstrates the predicted knee behavior of resistivity similar to samples L1-L5. We include here data on previously reported samples H1-H5 [23] to highlight the consistency in both the resistivity and positron data across these two sets of samples. The growth conditions of H1-H5 are similar to H6-H13 with minor fluctuations in precursors flow.”

R3: 2. 1. 99: The isosbestic point is not #2, but #3. #2 corresponds to the white line.

A: We thank the referee for spotting this typographical error, we have corrected it.

R3: 3. 1. 107: The text states, “Figure 3b shows the comparison of the experimental spectra obtained in samples H10 – H13 and L2 – L5.” However, it seems that only H13 is shown from the H series.

A: We thank the referee for spotting this, we have corrected the text to read:
“[...] samples H13 and L2 – L5.”

R3: 4. 1. 117: The manuscript states, “We note that there are no major differences in the simulated XANES spectra between the various Si_{Al}-Si_i complexes, except for Si_{Al}-Si_i.” However, if Si_{Al}-Si_i exhibits a major major difference, the same could be argued for Si_{Al}-H_i. While I agree with the statement that “\” (l. 120) and they may not be important, a more detailed analysis of the spectral differences would be beneficial.

A: We thank the reviewer for this suggestion. We have improved the discussion of the spectral features in the section on X-ray absorption near edge structure spectroscopy by adding the following text in the section on finite difference method near edge structure calculations:

“We note that most of simulated XANES spectra show similar behavior, except for Si_{Al}-H_i and Si_{Al}-Si_i.

In the simulated XANES spectra, interstitials near the Si atom and substitutional impurities occupying the nearest N sites contribute most significantly to the appearance of spectral features #4 and #5. Around 1847 eV, most simulated systems demonstrate common behavior, characterized by the presence of two peaks with variations in intensity and slightly also in energy. However, systems containing Si_{Al}-H_i and Si_{Al}-Si_i pairs deviate from this general trend. In Si_{Al}-H_i complexes, a high-intensity peak (#4) appears below 1847 eV, followed by a smaller peak (#5) above 1847 eV, with less than one-third of the intensity of feature #4. In contrast, in the Si_{Al}-Si_i system, feature #4 is less than 70% of the intensity of feature #5, and the latter is shifted approximately 1 eV to lower energies compared to all other samples. All simulated spectra exhibit a small peak at 1852 eV (features #7 and #8), with slight variations in centroid position and intensity.

The spectral features #9 and #10 are primarily influenced by the atomic species occupying the cation site nearest to the absorbing Si atom. For Si-complexes in AlN, all simulated spectra display a similar trend, with a peak appearing around 1860 eV, whose intensity varies depending on the composition of the system. Once again, Si_{Al}-H_i and Si_{Al}-Si_i systems appear as outliers. In the Si_{Al}-H_i system, the peak shifts slightly to 1859 eV. In Si_{Al}-Si_i system, there are three peaks at 1859, 1861, and 1863 eV characteristic to

the introduction of a Si atom. This unique behavior is exclusive to the $\text{Si}_{\text{Al}}\text{-Si}_i$ system, with the third peak corresponding to feature #11. For feature #12, all simulated spectra exhibit a low-intensity peak near 1867 eV. However, in the experimental spectra of H6–H9, where Si is predominantly in a Ga-rich environment, feature #12 is absent, and the spectra instead show a smooth transition to the continuum. We conclude that the slight differences between the various Si-related in AlN complexes are likely to induce broadening of the various features as observed in experiments, but they cannot account for the spectral differences between H6-H9 and each of the other groups, H10-H13 and L2-L5. We also note that the concentrations of O, H and C in the samples are significantly lower than the Si doping.”

R3: 5. 1. 142: 132 Al atoms should be 131 Al atoms assuming Si substitutes Al. Please verify.

A: We thank the referee for spotting this typographical error, it was corrected to 131.

R3: 6. 1. 170: Typographic error: t othe -> to the

A: Thanks, corrected.

R3: 7. How were the formation energies calculated? Was the usual Zhang-Northrup type formula used? Please clarify.

A: We follow the procedure outlined in the review by Freysoldt *et al.*, originally formulated by Zhang and Northrup.

We have modified the methods section in the main manuscript to read:

“We calculated the formation energy of a charged defect as the energy difference between the investigated system and the components in their reference states following the procedure outlined in the reviews by Van de Walle and Neugebauer [47], and Freysoldt *et al.* [21], originally formulated by Zhang and Northrup [48].

R3: 8. 1. 290: Was a Gamma-point-only sampling sufficient for convergence in the larger system? Has the convergence been checked? Given that the (+/-) transition levels are located very close to the conduction band minimum, even a subtle change in formation energies could be important.

A: We agree with the reviewer’s concern. We tested the convergence by the comparison of Gamma-point-only and with denser Gamma-centred \mathbf{k} -points meshes in 288 atom systems. We have modified the section on electronic structure theory in the main manuscript to read:

“The use of \mathbf{k} -points mesh containing Gamma point was tested for convergence against denser Gamma-centred \mathbf{k} -points meshes in 288 atoms systems.”

R3: 9. 1. 296: terminology “polarity”: The term polarity appears to be used in a context related to growth conditions. If this is the intended meaning, “polarity” may not be the most appropriate term. Please reconsider.

A: Good point. We have modified the text to read:

“We neglected the growth conditions (metal- or nitrogen-rich) of the systems and left the formation energy in the relative scale.”

R3: 10. Figure 4: The DX center appears unstable except in the case of 12 Ga in cluster Aside Cluster case. The (+/-) transition level seems to be above the gap, assuming that the band gap position is at the vertical dotted line. Have you calculated the effective correlation energy U for these states? Additionally, in the supplementary information 1. 66, U is used without definition. This should be explicitly defined.

A: Gordon *et al.* (Phys. Rev. B 89, 2014) suggests a simpler condition for DX center stability by observing the (+/-) transition level in relation to the conduction band. The (+/-) charge-state transition level is a thermodynamic level that determines the stability between positive and negative charge states of the defect (Van de Walle and Neugebauer, J. Appl. Phys. 95, 2004).

We modify the section on electronic structure calculations in the supplemental materials to read:

“The effective correlation energy U associated with the formation of DX^- is usually defined as

$$U = E^+ + E^- - 2E^0 \quad (2),$$

where E^+ and E^0 are the formation energies of the donor impurity in the positive and neutral charge states, and E^- is the formation energy in the negatively charged DX configuration. [6] This reaction describes the transition of two Si donor impurities in the neutral state into the positively charged donor and the impurity in the negatively charged DX configuration. Traditionally, negative U values indicate a stable DX center. However, since we are modelling the formation energies using alloy systems, it would make no sense to calculate all the energies in Eq. (2) using the same Si position and the same model with same configuration of Al and Ga, but one would have to consider how the distribution of formation energies would look like. This would require extensive work, but luckily there is a simpler solution. For a DFT cell with only one Si atom in either neutral, positive or negative DX states, the position of the (+/-) charge-state transition level in relation to CBM determines the stability between positive and negative charge states of the defect. [7]”

Manuscript: NCOMMS-24-80259

Title: Short-range order controlled amphoteric behavior of the Si dopant in Al-rich AlGaN

Comments by reviewers in blue color, and our answers are in black color.

Reviewer #2 (Remarks to the Author):

R2: Ga-rich regions due to random alloy fluctuations do not constitute short-range order. These regions are randomly distributed within the alloy and arise from statistical fluctuations. For instance, would such Ga-rich regions—emerging in an otherwise random alloy configuration—be reflected in the Warren-Cowley short-range order parameters?

A: We do not assume a deviation from the random alloy configuration of the Ga (and Al) atoms. We use the term “short-range order” to refer to the immediate local environment of Si atoms within the range of second nearest neighbour shell to be the determining factor for donor/acceptor behavior of Si which is directly evident from the differences in crystalline lattice around Si from the XANES experiments. Our data show that Si is preferentially incorporated next to (a high number) of Ga atoms, manifesting an ordering of Si with respect to Ga atoms as Si is not incorporated at random lattice positions. Hence our opinion is that the use of this term is appropriate: short range order (SRO for short) in crystalline alloys refers to the regular and predictable arrangement of atomic species over a short distance, usually with one or two atom spacings.

R2: Band alignment can be achieved or estimated by aligning w.r.t. the branch-point energy.

A: It is true that the branch point energy has been used for potential alignment in cases, in which it is hard to find a constant potential reference level using the macroscopic average of the potential. This is the case, for example when evaluating band offsets that involve semiconductor alloy systems, see e.g., Landmann et al. [Phys. Rev. B 95, 155310 (2017)]. Also, the branch point energy method enables one to estimate the alignments using affordable bulk calculations only.

In the present manuscript the alignment of the potential is related to the positions of the band edges of the surrounding, macroscopic AlGaN environment in each one of the panels in Fig. 4. The estimate of the magnitude of its band gap is based on linearly interpolating the HSE band gaps of AlN and GaN. A proper alignment relative to the band edges of the environment, no matter whether this is done using the branch point energy or vacuum level, would require a careful construction of a large number of physically relevant alloy models of sufficient size. This is a computationally demanding project and beyond the scope of the present work.